# The Impact of Culture-, Health- and Nature-Based Engagement on Mitigating the Adverse Effects of Public Health Restrictions on Wellbeing, Social Connectedness and Loneliness during COVID-19: Quantitative Evidence from a Smaller- and Larger-Scale UK Survey

**DOI:** 10.3390/ijerph20206943

**Published:** 2023-10-19

**Authors:** Linda J. M. Thomson, Neta Spiro, Aaron Williamon, Helen J. Chatterjee

**Affiliations:** 1UCL Arts & Sciences, University College London, London WC1 6BT, UK; 2UCL Biosciences, University College London, London WC1 6BT, UK; 3Centre for Performance Science, Royal College of Music, London SW7 2BS, UK; 4Faculty of Medicine, Imperial College London, South Kensington Campus, London SW7 2AZ, UK; neta.spiro@rcm.ac.uk (N.S.); aaron.williamon@rcm.ac.uk (A.W.)

**Keywords:** COVID-19, loneliness, public health restrictions, social connectedness, wellbeing

## Abstract

Numerous UK surveys conducted during COVID-19 examined the pandemic’s detrimental effects on health, and the consequences of lockdown and other public health restrictions on mental health. Some surveys considered specific populations and social inequities exacerbated during COVID-19. Fewer surveys examined the ways in which the adverse effects of public health restrictions, such as lockdown, shielding and social distancing, might be alleviated. Drawing upon self-determination theory, the purpose of the current study was to assess whether culture-, health- and nature-based engagement would mitigate the effects of these restrictions on psychological wellbeing, social connectedness and loneliness. Quantitative data from a smaller-scale survey (*n* = 312) and a subset of questions embedded in a larger-scale survey (*n* = 3647) were analyzed using univariate and multivariate methods. Frequency of engagement, whether participation was online or offline and with or without other people, and the extent to which type of participation was associated with psychological wellbeing, social connectedness and loneliness were examined. Sports and fitness, gardening and reading occurred frequently in both surveys. For the smaller-scale survey, increases in connectedness and frequency of participation and decreases in loneliness were significantly associated with improved wellbeing, whereas the type of participation and age range were not significant predictors. Outcomes from the smaller-scale survey approximated the larger-scale survey for measures of loneliness, type and frequency of participation and proportion of respondents in each age range. As the frequency of participation was a significant predictor of wellbeing, but the type of participation was not significant, the findings implied that any type of participation in a sufficient quantity would be likely to boost wellbeing.

## 1. Introduction

The COVID-19 pandemic and resultant public health restrictions yielded devastating effects on people’s lives; numerous UK surveys, for example, analyzed effects of the pandemic on health including symptom prevalence [1], transmission rates [2], new variant identification [3] and vaccine effectiveness [4,5]. Given concerns about the spread of the virus, severely reduced social contact due to public health restrictions, such as lockdown, shielding and social distancing and the subsequent closure of social services, it was expected that people’s mental health would suffer. Five days before the first UK lockdown (23 March 2020), the World Health Organization (WHO) identified that the crisis was ‘generating stress throughout the population’ [6] (p. 1). Consequently, there was a strong emphasis on surveys in the UK examining the effects of COVID-19 on mental health including, firstly, the impact on people with pre-existing mental illnesses [7,8,9,10,11]; secondly, the extent of common mental health disorder (anxiety and depression) arising from public health restrictions [12,13,14,15]; thirdly, mental health outcomes amongst specific sectors of the population [16,17,18,19]; fourthly, effects of the pandemic on psychosocial inequity [20,21]; and fifthly, ways in which the adverse effects of lockdown and other restrictions might be alleviated in terms of relieving loneliness and improving wellbeing through participation in culture-, health- and nature-based engagement [22,23,24,25]. The first four categories of survey illustrate the scope of research conducted during COVID-19 and assess deficits in mental health and wellbeing, whereas surveys in the fifth category, which align more closely with the current study, evaluate improvements in mental health and wellbeing derived from culture-, health- and nature-based engagement. The five categories will be addressed with a particular emphasis on the methods used and key outcomes.

For the first category, four UK surveys were conducted with young people (13–25 years) who had received mental health support prior to the onset of COVID-19 [7,8,9,10]. For the first survey during the initial lockdown (March 2020), when schools were closed to most children, 93% of over 2000 respondents agreed that public health measures had made their mental health worse. They were concerned about their grades; home learning, particularly where home environments were seen as difficult or dangerous; and lack of structure, pastoral support, routine and social connectedness [7]. The second survey, conducted when measures were announced to ease restrictions (June–July 2020), involved over 1000 young people who had accessed mental health support in the first three months of 2020. Around 81%of respondents agreed that public health measures had made their mental health worse, with 87% experiencing loneliness and isolation due to difficulties in maintaining social contact. Many reported a lack of motivation and purpose, concern for family members contracting the infection and returning to ‘normal’ with restrictions in place [8]. The third survey, conducted when schools re-opened (September 2020), focused on asking younger people (11–18 years), who had sought mental health support at some point in their lives, about their current mental health. Prior to returning to school, 58% described their mental health as poor or quite poor, but after returning, this figure increased by 11%. Some respondents reported feeling anxious or stressed about going back to school after many months away; they thought their mental health had started to improve during lockdown, only to deteriorate when returning [9]. The fourth survey, carried out during a new period of national lockdown (January–February 2021), included over 10,000 young people (13–25 years) who had needed mental health support since the beginning of the pandemic. Three quarters of respondents found the second lockdown harder to cope with, although 79% believed their mental health would improve once restrictions were lifted. Lockdown appeared to cause additional pressures of isolation, loss of routine and challenges in accessing mental health support identified in the three previous surveys [10]. In the same category, a longitudinal study explored the impact of COVID-19 on changes in mental health and wellbeing, service closures and social support for 377 respondents comprising older adults and people with dementia and their carers who completed an online or phone survey three times over 16 weeks [11]. Between the dates of this survey (April–August 2020), the first lockdown was extended before restrictions started to be eased with non-essential retail re-opening (June 2020), and although lockdown was eased further, restrictions such as shielding and social distancing remained in place. When social services closed during lockdown, many people were left without vital face-to-face support, and a lack of digital literacy created unequal access to online services [11]. The authors found that over time, anxiety reduced while depression increased for all respondents; mental wellbeing increased for older adults and people with dementia but not for their carers.

In the second category, the ‘COVID-19 Survey in Five National Longitudinal Studies’ was carried out during the first UK lockdown when the government announced a conditional plan for lifting restrictions (May 2020). The survey considered social isolation and mental health for 18,000 respondents who were tracked since childhood from five nationally representative cohorts, born in 1946, 1958, 1989–1990 and 2000–2002, participating in the ‘National Study of Health and Development’; ‘National Child Development Study’; ‘1970 British Cohort Study’; ‘Next Steps’; and ‘Millennium Cohort Study’, respectively [12]. The COVID-19 survey was designed to understand the economic, health and social consequences of COVID-19 and give insight into how people’s experiences might depend on their earlier lives. In addition to closed questions, an open-ended question asked respondents to describe their experiences of the pandemic. Preliminary themes included effects of social isolation, impacts on mental and physical health and concerns about loved ones, childcare, employment and finance [12]. A separate paper examining mental health over four generations showed generational differences among those experiencing poor mental health and loneliness during lockdown; anxiety and depression were more common among younger age groups with female respondents reporting worse mental health than male respondents, consistent with pre-pandemic interviews [13]. Respondents aged 19 years reported higher levels of loneliness (42% male; 45% female) on the UCLA 3-item Loneliness Scale [26] than those aged 30 years (26% male; 34% female) or 50 years (21% males; 27% females) [13]. Using the same five cohort studies, further research conducted during lockdown (May 2020) looked at relationship quality for couples and social support in households, specifically relationship satisfaction and conflict [14]. The survey showed that the oldest generation (74 years) was more likely to report decreased conflict with people around them (17.1%) compared with the youngest generation (19 years) who reported the most conflict (20.3%). Similarly, the highest satisfaction in relationships was reported by 74-year-olds, and the lowest satisfaction was reported by 19-year-olds. Within this second category, an international cross-sectional survey investigated the psychological impact of COVID-19 on emotions, behavior and wellbeing for over 7000 UK respondents, and a further 600 from Canada, Holland, India, Greece and Singapore [15]. The first wave of the survey (May–July 2020) measured depression, anxiety, impact, loneliness and social support. Healthcare professionals, constituting around half of the cohort, showed greater adverse psychological impact with higher proportions of mild depression and anxiety than those from the general population. The survey also found that older adults and women reported higher compliance with government advice whereas those with high alcohol or drug use, and prior suicidal thoughts reported lower compliance [15].

In the third category, surveys considering the mental health of specific sectors of healthcare professionals focused on medical doctors. The British Medical Association (BMA) ‘COVID-19 Tracker Survey’ with over 7800 respondents, for example, took place almost a year after the first lockdown (February 2021) [16]. When doctors were asked if during the pandemic they had suffered from any form of depression, anxiety, stress, burnout, or emotional distress associated with or worsened by their work or study, 51% agreed and 43% disagreed so responses were inconclusive, possibly due to a reluctance to suggest that their professional capacity might have been adversely affected. However, when respondents were asked about their current level of health and wellbeing compared with that during the first wave of COVID-19, only 14% thought it was slightly or much better; 34% thought it was about the same; but 51% thought it was worse or much worse [16], implying a recognition in hindsight of a wellbeing decline.

Along with mental health outcomes, research in the fourth category considered effects of the pandemic on psychosocial inequity, particularly whether there was a greater impact on those already disadvantaged pre-COVID, thereby widening inequalities. Using survey data from the ‘UK Household Longitudinal Study’ (UKHLS), researchers analyzed findings from the General Health Questionnaire (GHQ) prior to and during the first wave of COVID-19 (April–July 2020) to determine the level of socioeconomic inequality in psychological distress [20]. The authors found that the prevalence of psychological distress increased by around nine per cent (18.5–27.7%) between these dates, accompanied by a systematic increase in total inequality. However, measures of relative socioeconomic inequality did not increase as psychological distress was broadly diffused across the population. Aligning with the evidence on physical health, the results showed a worsening of overall GHQ levels during the first wave of the pandemic. Compared with pre-COVID, the specific dimensions affected were the ability to enjoy daily activities, playing a useful role, problems with concentration and unhappiness [20]. An update of the research, which extended the span of the UKHLS survey data on socioeconomic inequality in mental health to include the summer of limited restrictions and the second national lockdown (up to March 2021) [21], tracked the same respondents. Limited recovery of mental health (July–September 2020), with GHQ scores still higher than pre-pandemic, led to a significant decline (April 2020–March 2021). Demographic factors contributed most to inequality during the two lockdowns; a regression analysis showed that younger women (20–34 years) suffered most, and older men (65+ years) were the least distressed, acting to widen the gaps between age and gender groups. Highly affected items were related to decision making (September 2020), and depression, unhappiness and concentration (November 2020) [21]. These substantial decreases demonstrated that COVID-19 had a prolonged and damaging effect on the UK population’s mental health.

In contrast to assessing declines in mental health, as operationalized by the first four categories of survey, surveys in the fifth category, of which there are fewer, explored the mental health and wellbeing benefits of participation in culture-, health- and nature-based engagements during COVID-19, which is the focus of the current study. One possible reason for the relatively low number of surveys in this category is that, prior to 2020, UK surveys presented a relatively inconsistent picture of arts and cultural engagement. For example, several studies analyzed data from ‘Taking Part’, an annual survey of adults (16 years and above) and children (5–15 years) carried out by the Department for Digital, Culture, Media and Sport (DCMS) in England [27]. A study looking across three waves of Taking Part (2005–2006; 2008–2009; 2010–2011) showed that less than nine per cent of the population took part in any state-supported form of culture, and those who did take part tended to be privileged and well educated [28,29]. About half of the respondents had low levels of engagement with culture but took part in other activities such as ‘pubs, darts and gardening’, and 11 per cent were ‘detached from mainstream pastimes and social events outside of television’ [28] (p. 169) though may not have disengaged depending on the type of television programs viewed. Data analyzed from over 7000 respondents in Taking Part (2016–2017) aimed to identify whether participation in culture-, health- and nature-based engagement differed amongst those with varying levels of mental health. Findings showed no difference in participation amongst individuals experiencing high levels of anxiety but individuals experiencing low levels of happiness were less likely to engage in popular culture such as live music or cinema; high art such as opera or ballet; or literary events such as book fairs [30]. The DCMS commissioned researchers to undertake analysis of data from another annual survey, ‘Understanding Society’ (2013–2014) with 40,000 respondents to develop an evidence base on the wellbeing impacts of cultural engagement [31]. Findings showed that participation in libraries, sports and the arts was associated with higher wellbeing [32]. Again, prior to COVID-19, and presenting a more recent picture of arts engagement and mental health, the ‘Health, Economic, and Social Impact of the Arts’ (HEartS) survey (2018–2019) looked at cultural participation for over 5000 UK adults [33]. The authors examined correlations between arts engagement and measures of wellbeing and loneliness, finding that over 97% of respondents engaged in one or more arts activities at least once; reading for pleasure and listening to music were the most popular. 

Additionally, subsets of questions on arts engagement were asked within other surveys conducted pre-COVID-19. For example, the longitudinal ‘Whitehall II Study’ (also known as the ‘Stress and Health Study’) in England, which recruited a cohort of over 10,000 Civil Service employees in 1985, included a question on involvement in social indoor games such as cards, bingo and chess [34], and the ‘English Longitudinal Study of Ageing’ (ELSA), which surveyed non-institutionalized people aged 50 years and above at enrolment in 2002, incorporated questions on arts engagement. For the latter, cross-sectional data from two cohorts (2004–2005; 2014–2015) were analyzed from over 6000 respondents, with more than half of these also providing longitudinal data [35]. The authors distinguished between ‘receptive engagement’, art already created experienced by a listener or viewer, and ‘participatory engagement’ involving practical arts creativity [33] (p. 3). The study assessed associations between frequency of receptive engagement (visits to cinemas; museums; galleries; etc.) and loneliness in older adults using the 3-item Revised UCLA Loneliness Scale [26] and found a negative correlation between receptive arts and loneliness; visiting cultural venues every few months appeared to reduce loneliness. A limited number of surveys have explored arts and cultural participation during COVID-19 with a subset of questions included within a wider set of questions mainly concerning aspects of mental health. In a survey determining multidisciplinary priorities for mental health research, over 2000 respondents contacted through a mental health charity mailing list of stakeholders with lived experience completed an online survey during the first five weeks of lockdown (March–April 2020). The respondents were asked for their two biggest concerns and their coping strategies for maintaining health and wellbeing [25]. Though the survey was biased because it had four times as many respondents who were women than men and fewer young people under the age of 18, it was complemented by the more representative public Ipsos MORI’s Omnibus survey. In line with other surveys conducted in the early pandemic, concerns included anxieties associated with work, money, obtaining food, fear of the virus, keeping in touch, and becoming mentally unwell yet unable to access mental health services [25]. A further survey, the ‘COVID-19 Social Study’, carried out from the onset of lockdown, was an online weekly survey exploring issues associated with the pandemic [23]. Week ten was specifically concerned with arts engagement during lockdown. More than 19,000 respondents rated whether their engagement was less than, about the same as, or more than usual. Four categories of home-based arts engagement were identified comprising digital arts and writing; musical activities; crafts; and reading for pleasure [23]. The study highlighted the value of arts participation as a coping strategy during stressful situations. It showed that although regular engagers pre-pandemic continued to participate, lockdown created new opportunities for others to engage virtually including younger adults (18–29 years), and people who had lost employment or were concerned about contracting the virus [23].

Of the large-scale surveys outlined above, most were concerned with the impact of culture, incorporating measures of wellbeing, loneliness and social connectedness with an emphasis on mental health. The current study identifies with the DCMS Cultural White Paper definition that no longer depicts culture as simply ‘being familiar with a select list of works of art and architecture, but the accumulated influence of creativity, the arts, museums, galleries, libraries, archives and heritage upon all our lives’ and holds that ‘each community has its own culture—its own history, museums and traditions’ [36] (p. 13). The definition of impact was taken from the discussion document, ‘The Social Impact of the Arts’, as a dynamic concept that assumes a cause-and-effect relationship and is measured through evaluation outcomes [37]. Similarly, in an essay on the cultural impact of museums, cultural impact is described as an effect and is distinct from cultural value, which is concerned with worth and importance [38]. The current study also considers the broader concept of cultural impact that ‘manifests in relation to the sense that people make of the world’ [38] (p. 12). A systematic review of cultural impact assessment found a divergent understanding of the term, cultural impact, in the cultural (funded arts and heritage) sector that includes the impact on culture and the impacts of cultural activities or interventions on individuals, and which involves both tangible and intangible assets [39]. The definition of health impact in the current study aligns with ‘both positive and negative changes in community health that are attributable to a policy, program or project’ [40] (p. 648). Conversely, in connection with a definition of community engagement, the WHO referred to community engagement as the ‘process of developing relationships that enable stakeholders to work together to address health-related issues and promote well-being to achieve positive health impact and outcomes’ [41] (p. 3). For nature-based impact, the current study draws upon the recent definition of nature-based interventions as ‘programs, activities or strategies that aim to engage people in nature–based experiences with the specific objective of enhancing health and wellbeing’ that can ‘facilitate change through a relatively structured promotion of nature-based experiences’ [42] (p. 2).

Given the national and local lockdowns restricting freedom of movement and public health measures to limit social contact, the aims of the current study were to determine how people spent their free time during COVID-19 and the impacts of these activities on psychological wellbeing, social connectedness and loneliness. The study drew upon self-determination theory (SDT) as a basis for the research [43]. SDT utilizes the concept that psychological needs are innate and ‘essential for ongoing psychological growth, integrity, and wellbeing’ and that there is a ‘fundamental human trajectory toward vitality, integration, and health’ [43] (p. 229). This theory of human motivation emphasizes the importance of satisfying the psychological need of ‘relatedness’ [43] (p. 228) or ‘connectedness’ [41] (p. 29), in addition to competence and autonomy, to maintain a sense of wellbeing. For the purposes of this study, the notion of relatedness or connectedness was defined as ‘the number of friends and frequency of interactions with family and friends and activities of societal value’ [44] (p. 361). Loneliness, in contrast, was defined as the ‘state of being without any company or in isolation from the community or society’ [45] (p. 526) and as a ‘subjective phenomenon… not necessarily synonymous with objective isolation, so that people can be alone without being lonely’ [46] (p. 32). SDT further asserts that, in addition to relatedness, it is necessary to satisfy the needs of autonomy and competence to achieve psychological health and wellbeing. Autonomy was seen to concern the experience of ‘integration and freedom’, whereas competence was thought to ‘energize human activity’ [43] (p. 231). For the current study, wellbeing was defined as ‘the dynamic process that gives people a sense of how their lives are going through the interaction between their circumstances, activities and psychological resources or mental capital’ [47] (p. 3). SDT hypothesizes that optimal development and wellbeing will be achieved under conditions that support the satisfaction of needs, whereas conditions that threaten or deprive the satisfaction of needs will diminish wellbeing [43]. Satisfaction could occur through engaging in a variety of behaviors differing between individuals and cultures.

Using evidence from a smaller- and a larger-scale survey, the objectives were to identify the lockdown behaviors reported in both surveys and assess the frequency of occurrence of these activities; determine how they were carried out (online alone; online with other people; offline alone; or offline with other people); and ascertain whether the type of participation was associated with the respondents’ feelings of social connectedness and level of loneliness. Participation alone was defined as ‘predominantly solitary activities’ such as reading and writing, whereas participation with others was defined as ‘predominantly social activities’ such as attending films and concerts [33] (p. 20). With respect to online and offline participation, few studies have examined differences in the effects of these types of engagement. A study examining the roles of online and offline features in sustaining virtual communities, for example, drew upon the technology acceptance model, which is a general model of user acceptance associated with new technologies, as the theoretical underpinning for their research [48]. The author found that perceptions of usefulness and ease of use had significant effects on feelings of belonging. Offline activities, however, also positively affected connectedness, suggesting that ‘online social ties among members cannot be sustained without strong offline interactions’ [48] (p. 132). A study of online searches for tourism information drew upon a user-centered human-computer interaction approach involving interactions with information environments. The authors found that barriers to online searching were slow internet connection speeds, a lack of patience in waiting for pages to load and out-of-date information, which often led to an alternative offline solution such as making a phone call [49].

Aligning with SDT [43], the current study hypothesized that improvements in wellbeing would be associated with increases in feelings of connectedness and decreases in feelings of loneliness. It was further hypothesized that improvements in wellbeing would be associated with increases in the frequency of engagement as ‘the degree to which people are able to satisfy their basic psychological needs’, which is a critical issue in the self-determination theory [43] (p. 227). Based on previous research, however, it could not be predicted how the way in which participation was carried out might affect these outcomes. 

## 2. Materials and Methods

The study used univariate and multivariate methods to analyze data generated by the smaller- and larger-scale surveys. The ten-question, smaller-scale ‘Community COVID: What have you been doing?’ (Community COVID) survey comprised three open questions followed by seven closed questions (Section A.1). Community COVID questions were embedded in the larger-scale ‘Health, Economic, and Social Impact of the Arts’ Phase 2 Engagement survey (HEartS Engagement) to increase the reach of data collection by including both a greater number and a wider cross-section of respondents; the larger survey used quota sampling, whereas the smaller survey used convenience sampling. Seven of the Community COVID questions were the same as those in the HEartS Engagement ‘Participatory Arts’ section, four were the same as those in the ‘Receptive Arts’ section, and two questions formed a new section, Culture-, health-, and nature-based engagement’ (Section A.2), not included in the original HEartS (2018–2019) Phase 1 survey [33]. Community COVID asked respondents one overall question (Question 5) about how they normally took part in the activities they had listed but for HEartS Engagement, the same question was asked for each set of activities. Consequently, more Hearts Engagement than Community COVID questions were used to cover the same topics. 

Community COVID and HEartS Engagement surveys used the UCLA 3-item Loneliness Scale [26] to assess loneliness, a measure that was repeatedly demonstrated to display satisfactory internal consistency, test-retest reliability and concurrent and discriminant validity [26]; the 3-item short-form version used here is the recommended loneliness measure at the UK national level [50]. Community COVID additionally used the UCL Museum Wellbeing Measure [51,52] to assess psychological wellbeing, which was established using Cronbach’s alpha, producing an alpha level of 0.81, where scales of 0.7 and over are regarded as reliable [53]. Quantitative data comprising 8 Community COVID questions and 13 HEartS Engagement questions were analyzed using descriptive and inferential statistics in IBM SPSS v.25.3. Regression analysis was used to ascertain criterion predictive validity for the analysis of wellbeing and loneliness, and other constructs (how often respondents take part, how they take part and how connected they feel) were used to examine the extent to which each measure predicted other measures.

Both surveys used samples of UK adult participants aged 18 years and above from across the UK with approximately equal numbers of men and women (Table A1). Community COVID respondents (*n* = 312) were recruited anonymously through mailing lists of arts-, heritage- and nature-in-health partner organizations and were not paid for their participation. For the HEartS (2018–2019) Phase 1 survey (*n* = 5338), quotas were set for gender, age, geographical region, ethnicity, and education following the distribution of variables in the UK 2011 Census [54]. Respondents from this survey were recruited for HEartS Engagement Phase 2 (*n* = 3647) and paid a modest sum for their participation. Participant age range, UK region, and employment were recorded; age in years for HEartS Engagement was converted to age range to align with Community COVID. For both surveys, the median age range was 50–59, the most frequently occurring region was South East England, and fewer than five per cent of participants worked as professional artists, performers, or makers.

Ethical approval was obtained for the research. Links to Community COVID were sent via partner mailing lists, and links to HEartS Engagement were sent to the HEartS Phase 1 respondent pool. For both surveys, participants were asked to read the participant information and privacy notice and sign the consent form. Surveys including briefing documents were conducted online; Community COVID for 12 weeks (7 September–27 November 2020) and HEartS Engagement for four weeks (5–30 October 2020). Community COVID was collected on Opinio 7.12 and HEartS Engagement on the Qualtrics online platform. During this time, gatherings of more than six people in England were banned (from 14 Sept 2020); there was a return to working from home (22 Sept 2020); and the second national lockdown in England took place (5 November–2 December 2020) [55].

## 3. Results

The data from Questions 1 and 4–10 were analyzed using multiple regression, analysis of variance (ANOVA), *t*-tests, Chi-squared test and correlation and descriptive statistics. Questions 2 and 3, requiring qualitative responses, were omitted from these analyses. The Community COVID data for the frequency of activities (Question 1), frequency of participation (Question 4), type of participation (Question 5), connectedness with others (Question 6), wellbeing (Question 7), loneliness (Question 8), age range (Question 9) and UK region (Question 10) were analyzed separately. Omitting the frequency of activities and region, the above variables were entered into a regression model to determine predictor variables for wellbeing. An additional analysis of the Community Covid data with the HEartS Engagement data was carried out where variables were comparable, for the type of participation, connectedness with others, loneliness and the numbers of respondents in each age range and region.

### 3.1. Analysis of the Community COVID Survey

#### 3.1.1. Community COVID Questions

Question 1: ‘What have you done with your free time during COVID-19 restrictions?’ Respondents listed their main activities (*n* = 1639) and mean number of activities per respondent was calculated (mean = 5.25). Activities that occurred twice or more were grouped into 30 categories using similar wording to the (2018–2019) Phase 1 HEartS survey [33] wherever possible, and frequency and percentage of occurrence were calculated (Table A2). The highest frequency of response was for sports/fitness listed by 86% of respondents; other activities in the upper quartile comprised crafts, textiles or decorative arts (67%); gardening/looking after plants (53%); painting, drawing, printmaking, sculpture, etc. (41%); volunteering and community activities (32%); reading as a pastime (24%); cooking and baking (23%); and talking to friends and family/socializing (21%).

Question 4: ‘How often have you taken part in these activities compared with before COVID-19 restrictions?’ The respondents ticked one box from the seven possible responses (Table 1), where frequency refers to how frequently each response occurred, i.e., the number of respondents who selected each response with the percentage of respondents calculated from the total number of respondents; 83% of respondents took part more often (sum of a little more: 21%; quite a lot more: 37%; much more: 25%).

Question 5: How do you usually take part in these activities?’ The respondents ticked one box, and the frequency and percentage of the responses were calculated in the same way as in Question 4 (Table 1); 78% of respondents took part offline (offline alone: 42%; offline with other people: 36%). The data from the type of participation were entered into the regression analysis (3.1.2).

Question 6: ‘How much have these activities helped you feel connected with other people?’ The respondents ticked one box from the response items, including ‘Not at all’; ‘A little’; ‘Around half the time’; ‘Often’ and ‘Always’, and the frequency and percentages of the responses were calculated in the same way as in Question 4 (Table 1); 63% of the respondents felt connected with other people half of the time or more.

Question 7: ‘Please select one box (out of five) for each word to indicate how you feel on a typical day’. The respondents completed the Museum Wellbeing Measure rating, with each of the six items between 1 and 5 giving a total wellbeing score between 6 and 30, and the means and standard deviations (SD) were calculated (Table 2). The mean total wellbeing (17.98) was below the cut-off point of 18 for high wellbeing. A repeated measures ANOVA of the wellbeing items (active; alert; excited; enthusiastic; happy; inspired) showed highly significant differences between the items, F(5,1555) = 52.68, *p* < 0.001; Bonferroni *t*-tests showed that the item ‘excited’ was significantly lower than the other five items, t(311) = 8.70, *p* < 0.001 for the smallest difference (Figure 1).

Question 8: ‘Please select one box (out of three) for each question to indicate how you feel on a typical day’. The respondents completed the UCLA 3-item Loneliness Scale and means and standard deviations were calculated (Table 2). The mean total loneliness (5.27) was below the cut-off point of six for high loneliness. A repeated measures ANOVA of the loneliness items (How often do you feel that you lack companionship? How often do you feel isolated from others? How often do you feel left out?) showed highly significant differences, F(2,622) = 18.68, *p* < 0.001; the Bonferroni *t*-tests showed that ‘How often do you feel left out?’ was significantly lower than the other two items, t(311) = 4.18, *p* < 0.001 for the smallest difference (Figure 2).

Question 9: ‘Which age group are you in?’ The respondents selected one out of seven age groups. All age groups were represented by at least 14 respondents; over 60% were older adults aged 60–69 (25%), 50–59 (21%) and 70–79 (20%).

Question 10: ‘Where do you live?’ Each of the 14 UK regions were represented by a minimum of three respondents; over 50% were from southern England: South East (26%), London (14%) and South West (13%). The means and standard deviations were calculated for wellbeing (out of 30) and loneliness (out of 9) ratings across the regions. The wellbeing ratings were the highest in the East Midlands (m = 20.00, SD = 6.00) and Northern Ireland (m = 19.80, SD = 7.70), and lowest in North Wales (m = 14.70, SD = 1.60) and South Wales (m = 16.10, SD = 4.10) whereas the loneliness ratings were the lowest in the South West (mean = 4.80, SD = 1.75) and Northern Ireland (m = 4.85, SD = 2.25), and the highest in the East of England (mean = 6.16, SD = 3.15) and North Wales (m = 5.95, SD = 1.55) (Figure 3).

#### 3.1.2. Predictor Variables for Wellbeing

A multiple regression analysis was conducted with predictor variables of loneliness (interval level data), age range, frequency of participation, and connectedness with others (ordinal level data), and participation offline or offline and alone or with others (dichotomous categorical data), and criterion variable of wellbeing (interval level data) (Table 3). A significant model emerged (F(6,305) = 25.26, *p* < 0.001), which explained 32% of the variance in the measure of wellbeing (adjusted R^2^ = 0.32). The regression coefficients for the predictor variables entered into the model showed that connectedness was a highly significant predictor with a positive relationship to wellbeing, where increased connectedness was associated with increased wellbeing (*p* < 0.001); frequency of participation was a significant predictor with a positive relationship to wellbeing where increased frequency was associated with increased wellbeing (*p* < 0.05); loneliness was a highly significant predictor with a negative relationship to wellbeing where decreased loneliness was associated with increased wellbeing (*p* < 0.001). The age range and participation, whether offline or online, or alone or with others, were not significant predictors of wellbeing (*p* > 0.05). There was a significant correlation, however, between age range and loneliness (r = 0.13, *p* < 0.001), where respondents aged 70 years and over were less lonely than the other age groups. There were significant correlations between connectedness and type of participation, both alone or with others (r = 0.338, *p* < 0.001), or offline or online (r = 0.250, *p* < 0.001). The type of participation showed an interaction where there was a greater difference between being alone and with others for offline participation than online participation. As it was inappropriate to use a parametric test, a Chi-squared test, which is suitable for frequency counts, was conducted, which indicated a significant difference (Chi^2^ = 5.66; *p* < 0.02); significantly more respondents participated offline alone than offline with others or online.

### 3.2. Analysis of Both Surveys

#### 3.2.1. Type of Participation and Connectedness with Others

These sections can be compared, as both surveys asked the respondents how they participated and to rate the extent to which their engagement in activities helped them feel connected with other people, though this was achieved using slightly different methods. In Question 1, the Community COVID survey asked the respondents to list their activities, Question 5 asked how they had taken part in them (online alone; online with others; offline alone; or offline with others) and Question 6 asked the respondents to rate their connectedness. The HEartS Engagement survey asked the respondents to tick a list of 20 engagement activities to indicate which ones they had participated in within the previous month and how they had taken part in them (online alone; online with others; offline alone; or offline with others) (Table A3). To align the HEartS Engagement data with the Community COVID data, the respondent numbers for each of the four types of participation were calculated from each of the 20 HEartS Engagement activities. For HEartS Engagement, however, 1488 respondents selected ‘not applicable’, leaving data from 2159 respondents which was entered into the analysis (Figure 4).

#### 3.2.2. Loneliness

For both surveys, the means and standard deviations were calculated for each of the UCLA 3-item Loneliness Scale questions (Table 4). As the sample sizes between the surveys were unequal, violating assumptions of variance for ANOVA, four independent *t*-tests were conducted to compare the Loneliness Scale items and the total scores across both surveys. For ‘How often do you feel that you lack companionship?’, t(311) = 2.61, *p* < 0.01; for ‘How often do you feel isolated from others?’, t(311) = 2.30, *p* < 0.02; for ‘How often do you feel left out?’, t(311) = 0.07, *p* < 0.95; and for the total scores, t(76) = 0.13, *p* < 0.90. The Bonferroni correction, used to control for multiple comparisons, showed that the only significant difference between the two surveys was for ‘How often do you feel that you lack companionship?’, where the responses for Community COVID were significantly greater than for HEartS Engagement (*p* < 0.04). The findings were therefore statistically similar for both surveys with the exception of the companionship question. For both surveys, the mean rating for ‘How often do you feel left out?’ was the lowest, and the mean rating for ‘How often do you feel isolated from others?’ was the highest. Mean ratings for total loneliness for Community COVID (m = 5.27) and HEartS Engagement (m = 5.01) were both below the cut-off point for high loneliness. The ratings were plotted for each item so that the surveys could be compared for the three response options (Hardly ever; Some of the time; Often) (Figure 5). Bonferroni *t*-tests were used to compare pairs of response options across both surveys.

#### 3.2.3. Correlation of Respondents

A Spearman correlation of numbers of respondents in each region from both surveys showed a highly significant positive correlation; rho = 0.765, *p* < 0.002, one-tailed. A Spearman correlation of numbers of respondents in each age range from both surveys showed a weaker positive correlation that came close to significance; rho = 0.613, *p* < 0.07, one-tailed. The correlations indicated that although numbers of respondents for Community COVID were less than nine per cent those of HEartS Engagement, the smaller sample approximated the larger sample in terms of number of respondents in each age range and region.

## 4. Discussion

The aims of the current study were to establish how participants spent their free time under lockdown and other public health restrictions, and evaluate the effects of these activities. The objectives were to analyze the data from both surveys to assess the frequency of activities and determine how these were carried out (online or offline; with or without other people). The purpose of the study was to assess whether culture-, health- and nature-based engagements during COVID-19 would mitigate the effects of public health restrictions on psychological wellbeing, social connectedness and loneliness. In line with the hypotheses and in keeping with SDT [43], the respondents who felt more connected or related to others reported better psychological wellbeing than those who felt less connected. Around four fifths of the Community COVID respondents reported feeling connected to other people always or often, though this was double that for HEartS Engagement respondents. Less than a tenth of the Community COVID respondents felt not at all connected, whereas for HEartS Engagement, the number was twice as many. For Community COVID, loneliness was inversely associated with wellbeing; the respondents who reported higher levels of loneliness experienced lower levels of wellbeing, whereas the respondents with lower levels of loneliness experienced higher levels of wellbeing. As HEartS Engagement did not measure wellbeing directly, correlations with loneliness and the frequency of participation could not be carried out. For Community COVID, an increase in the frequency of participation was significantly associated with improved wellbeing, but as the type of participation was not a significant predictor, the study’s theoretical framework was scrutinized to explain this finding. SDT utilizes the concept of innate psychological needs to differentiate outcomes from the processes by which these outcomes are achieved. Here, the processes involved were whether participation was carried out online or offline and alone or with others, and consequently, how participation occurred may have had less impact than the outcomes and frequency of that participation. SDT also proposes that the needs of autonomy and competence should be satisfied in addition to the need of relatedness for wellbeing to occur [43]. Although the current study did not measure autonomy and competence directly to examine their effects on wellbeing, anecdotal evidence from the survey comments suggested that most participants felt that they needed to be independent or self-sufficient during lockdown, and in terms of competence, they pursued activities that they already knew how to do and had the necessary equipment for, indicating a certain degree of proficiency. Consequently, with connectedness in place, and hypothetically, with autonomy and competence also present, SDT would predict that wellbeing would occur regardless of the processes involved in participation.

Community COVID’s wellbeing improvement findings align with those from the Understanding Society survey, which found that cultural participation such as in sports and the arts was associated with higher wellbeing [32], and the HEartS (2018–2019) Phase 1 survey, where arts engagement was associated with lower levels of loneliness and higher levels of wellbeing and social connectedness [33]. In contrast, HEartS Phase 1 authors found a positive association between greater arts engagement and depression and intense loneliness for those who were most highly engaged [33], which was not evident in Community COVID. In fact, the respondents from neither of the current surveys showed exceptionally high scores for loneliness in that they mostly stayed under or around the cut-off point for high loneliness set by the authors [26]. The question ‘How often do you feel isolated from others?’ received the highest ratings from both Community COVID and HEartS Engagement respondents. For Community COVID, the respondents were less likely to feel left out than isolated or lacking companionship. As with loneliness, the Community COVID respondents did not have particularly high wellbeing scores in that, again, they stayed around or below the cut-off point for high wellbeing set by the authors [51,52]. Out of the mood items for wellbeing, the respondents were significantly less likely to feel excited than active, alert, happy, enthusiastic or inspired. The findings of feeling isolated, only having a medium level of wellbeing and little excitement were not surprising given the devastating effects of the pandemic at the time of the survey that precipitated a return to working from home and the second national lockdown in England.

Of the activities reported for Community COVID, the highest proportion of respondents took part in health-based engagement in the category of sports and fitness with walking the most popular; for HEartS Engagement, this category was the sixth highest. Findings aligned with the Young Minds Survey 4 where half of the respondents reported that exercise such as walking or running was helpful for their mental health during lockdown [10]. The importance of exercise was also demonstrated in the COVID-19 Social Study which, in addition to other issues arising from the pandemic, explored the extent of physical activity through analysis of longitudinal data from almost 36,000 respondents in England for up to 22 weeks [22]. The period encompassed lockdown followed by easing of restrictions to allow unlimited outdoor exercise (May 2020) and reopening of outdoor gyms, playgrounds, and swimming pools (July 2020). Results showed that the number of respondents who exercised for three plus hours a week, increased in the first ten weeks and then decreased and stabilized, with less than a tenth showing an upward trend in physical activity and those who did not take any exercise steadily increasing [22].

Culture- and nature-based engagement also occurred in the upper quartile of responses for Community COVID and included crafts, where quilting and patchwork were prevalent, followed by gardening and looking after plants; painting and drawing; volunteering and community activities; reading as a pastime; cooking and baking; and talking to friends and family and socializing. Mental health research conducted during the pandemic found similarly that the main strategies for maintaining wellbeing were staying connected with family and friends, and volunteering and helping others; keeping busy with hobbies, crafts, art, music, reading, film, television, and home improvements; physical activity including walking, running, online exercise classes, accessing nature and the outdoors; and staying calm through, for example, mindfulness and meditation [24]. For HEartS Engagement, the upper quartile of respondents selected watching film or drama the most frequently followed by reading; gardening; listening to music and crosswords. For both surveys, the emphasis on cultural-based engagements was comparable with the Young Minds Survey 4 where half of the respondents suggested that listening to music was helpful, and a third that watching films was helpful [10]. For nature-based engagement, gardening and looking after plants was the most popular category, with visiting natural environments being listed less frequently. For some activities, particularly health-based engagement, it was difficult to separate the natural context from the healthy activity such as walking, cycling or free swimming in natural environments. Furthermore, the respondents were asked to list the activities they had taken part in rather than the environment in which the activity took place. The People and Nature Survey found that pandemic restrictions allowed them to find new ways to connect with nature, and two-thirds of respondents reported that they were taking more time to engage with nature on a daily basis [56]. However, in keeping with the current study, a quarter of the respondents had not spent any time in green spaces, and this figure was higher for lower-income households, maintaining pre-COVID social inequalities in terms of access to nature.

Both surveys showed that practicing or performing a play, or dance were in the lower quartile of activity categories. Data from the Understanding Society survey (2010–2012) suggested that only a tenth of the UK population engaged with participatory performing, visual, and literary arts [57], although Community COVID found that visual arts including painting and drawing occurred in the upper quartile. For the HEartS Phase 1 (2018–2019) conducted pre-COVID, reading as a pastime, and listening to music were the most popular activities [33]. The authors split engagement into three clusters: ‘low engagers’ who read occasionally; ‘receptive consumers’ who read and listened to music frequently and engaged with popular receptive arts such as cinema, concerts, and exhibitions; and ‘omnivores’ who frequently engaged in almost all arts activities [33] (p. 21). A third of activities for both current surveys were receptive arts and for Community COVID, a third were participatory arts. For the HEartS Engagement questions analyzed in the current study, participatory arts were selected around a sixth of the time, suggesting that Community COVID respondents could be classed as ‘omnivores’, but HEartS Engagement respondents as ‘receptive consumers’. For HEartS Engagement, reading as a pastime, and listening to music occurred frequently within the upper quartile in keeping with HEartS Phase 1 quite possibly because participants were a subset of those recruited through Qualtrics to participate in the original survey, though reading also occurred frequently for Community COVID respondents recruited through arts-, heritage- and nature-in-health partners. 

As in the ELSA survey [35] and the HEartS Phase 1 survey [33], the current surveys distinguished between participatory and receptive arts engagement, with HEartS Engagement including specific sections for these categories. The DCMS also distinguished between two sorts of arts activities though their categories were participatory arts and art events [31]. The Understanding Society survey showed that for 2013–2014, three quarters of adults in England participated in at least one arts activity and nearly two thirds attended at least one arts event [58]. For the same time period, over a third of adults engaged in both participatory arts and arts events [31]. In-person events however were banned during lockdown so these would have involved virtual attendance and, given that offline engagement was more prevalent than online, it is likely that most activities would have therefore been participatory. Although Community COVID did not have separate sections for participatory and receptive arts because respondents were asked to list their main activities as free text, over 90 per cent of activities reported were participatory in format, with arts engagement forming a third of these.

Given the inconsistencies between past surveys of leisure time activities (see Introduction) it was expected that the study would find differences between the Community COVID and HEartS Engagement surveys. It was not expected, however, that there would be commonalities between the surveys, particularly gardening and reading which were in the upper quartile of both responses and the category of sports and fitness which was at the top of the upper quartile for Community COVID and just below the upper quartile for HEartS Engagement. The Taking Part survey found high participation in reading for pleasure involving two-thirds of respondents, and in fitness activities involving a sixth of respondents [28], aligning with both current surveys. For Taking Part, greater numbers participated in informal leisure pursuits such as spending time with friends and eating out [28], and although face-to-face contact would not have been possible with people outside of the household during lockdown, Community COVID found that talking to friends and family were important aspects of maintaining social connections in line with the Young Minds Survey 4 where keeping in contact with friends and family was a key coping strategy for maintaining mental health [10]. Another important commonality between the current surveys was type of participation; offline occurred more frequently than online, and offline alone occurred more frequently than offline with others. For Community COVID, just under four-fifths of respondents took part offline rather than online and slightly more of these participated alone than with others. Similarly for HEartS Engagement, two-thirds of respondents took part offline, though twice as many participated alone than with others. It was therefore difficult to assess whether the perceptions of usefulness or the ease of use implicated the Technology Acceptance Model [48], with fewer respondents opting for offline engagement compared to online engagement.

### Strengths and Limitations

The Community COVID and HEartS Engagement surveys represent two of several surveys that assessed the impact of culture-, health- and nature-based engagement on mitigating the adverse effects of pandemic restrictions through improving wellbeing and connectedness and alleviating loneliness. This study adds generalizability to the existing body of literature by assessing these variables under the unique situation of COVID-19 public health restrictions, particularly for connectedness, maintaining consistency with SDT. Key findings from both surveys were comparable to a large extent despite the number of respondents for HEartS Engagement being more than ten times larger than for Community COVID. Commonalities between surveys offer testimony to their reliability and validity. Community COVID was only compared, however, with a subset of HEartS Engagement responses, the new section on culture-, health- and nature-based engagement, and a limited number of questions from receptive and participatory arts sections. Furthermore, nine activity categories from Community COVID were not included in HEartS Engagement as the latter solely concerned arts engagement, consequently, parallels drawn between the two surveys should be interpreted with caution. Although parallels can be drawn from the two surveys in terms of loneliness and social connectedness, psychological wellbeing could not be assessed across both surveys as HEartS Engagement did not include a wellbeing measure. Although there is indirect evidence from previous studies of mental health [12,13,14] indicating that wellbeing might have worsened during lockdown due to isolation and the absence of participation in engagement activities, there is no direct evidence of this for the respondents in the current study. Future surveys might consider using randomized controls where half of the participants are asked to refrain from engaging in activities so that wellbeing measures can be compared with those who do engage in activities. Furthermore, it should be noted that findings from the Community COVID survey represent a cross-sectional ‘snapshot’ in time during the COVID-19 restrictions; as this was not a longitudinal study, an analysis was unable to determine whether the outcomes would be sustained. 

## 5. Conclusions

In line with the purpose of this study, it was found that culture-, health- and nature-based engagement activities carried out during COVID-19 mitigated the effects of public health restrictions; Community COVID found that social connectedness, frequency of participation and loneliness were significant predictors of psychological wellbeing in keeping with SDT [43]. Increased connectedness and increased frequency of participation were associated with increased wellbeing, though the type of participation (online or offline; alone or with others) was not a significant predictor. Conversely, decreased loneliness was associated with increased wellbeing, although age range was also not a significant predictor of wellbeing despite loneliness being often associated with older age groups. In combining evidence from the smaller-scale Community COVID survey with a subset of questions from the larger HEartS Engagement survey, the current study determined how people spent their free time under lockdown and other restrictions associated with the later part of 2020. For Community COVID, over four-fifths of respondents took part in activities more often than pre-COVID, with the most frequently occurring activity being sports and fitness. Although the most frequently occurring activity for HEartS Engagement was watching films or dramas, a high proportion of respondents from both surveys selected gardening, reading, and sports and fitness as main activities. Consistent with previous research on social and cultural impact [37,38,39,40], an evaluation of the effects of a wide range of activities undertaken during pandemic restrictions showed that engagement appeared to improve psychological wellbeing, reduce loneliness and engender feelings of social connectedness. Perhaps the most interesting outcome for Community COVID was that the frequency of participation was a significant predictor of wellbeing, but the type of participation was not a significant predictor, implying that any form of engagement in a sufficient quantity would be likely to boost wellbeing.

## Figures and Tables

**Figure 1 ijerph-20-06943-f001:**
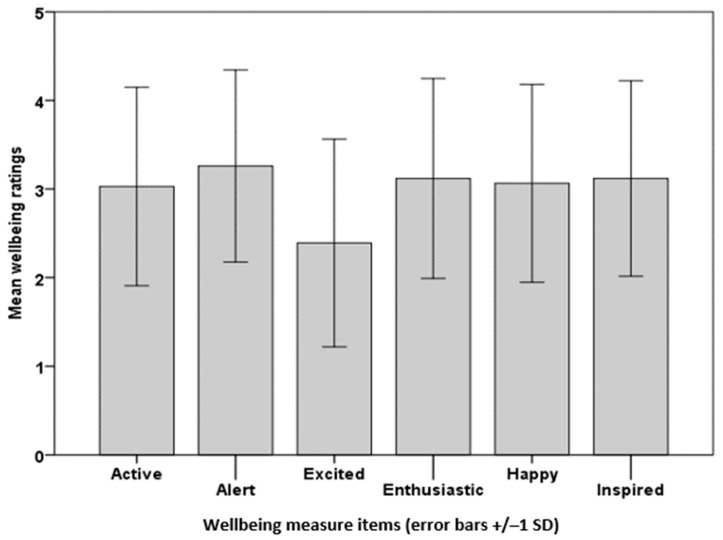
Community COVID: mean ratings of wellbeing scores.

**Figure 2 ijerph-20-06943-f002:**
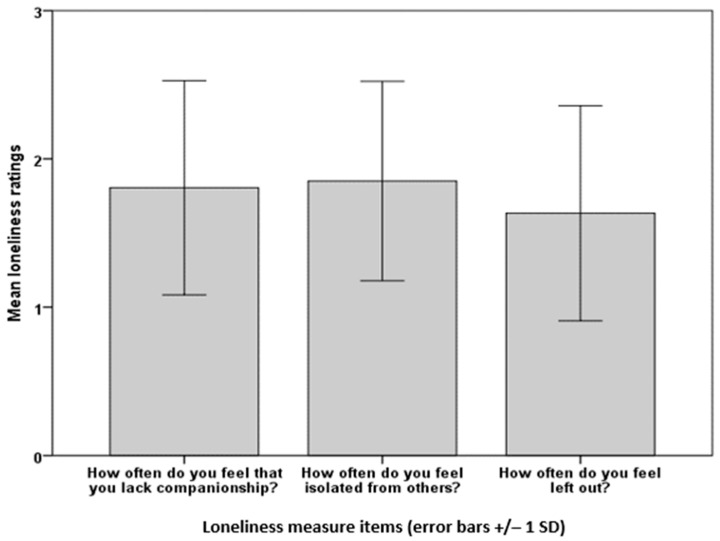
Community COVID: mean ratings of loneliness scores.

**Figure 3 ijerph-20-06943-f003:**
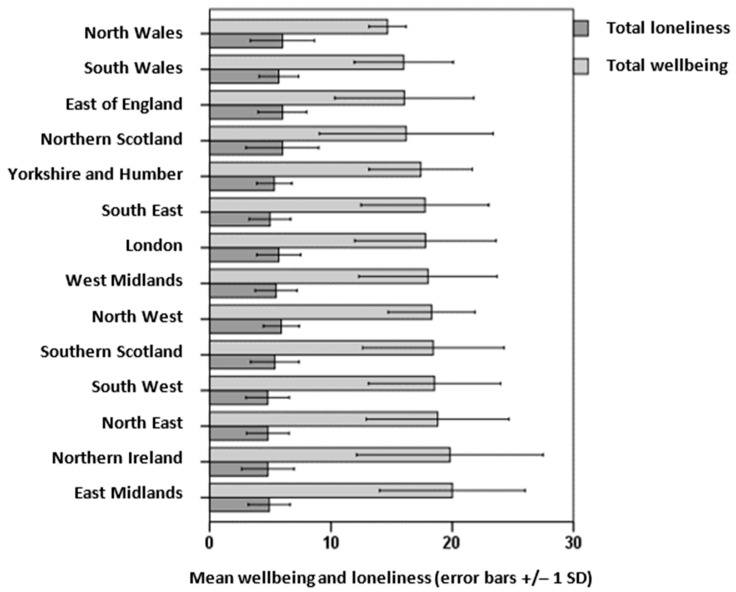
Community COVID: wellbeing and loneliness scores across UK regions.

**Figure 4 ijerph-20-06943-f004:**
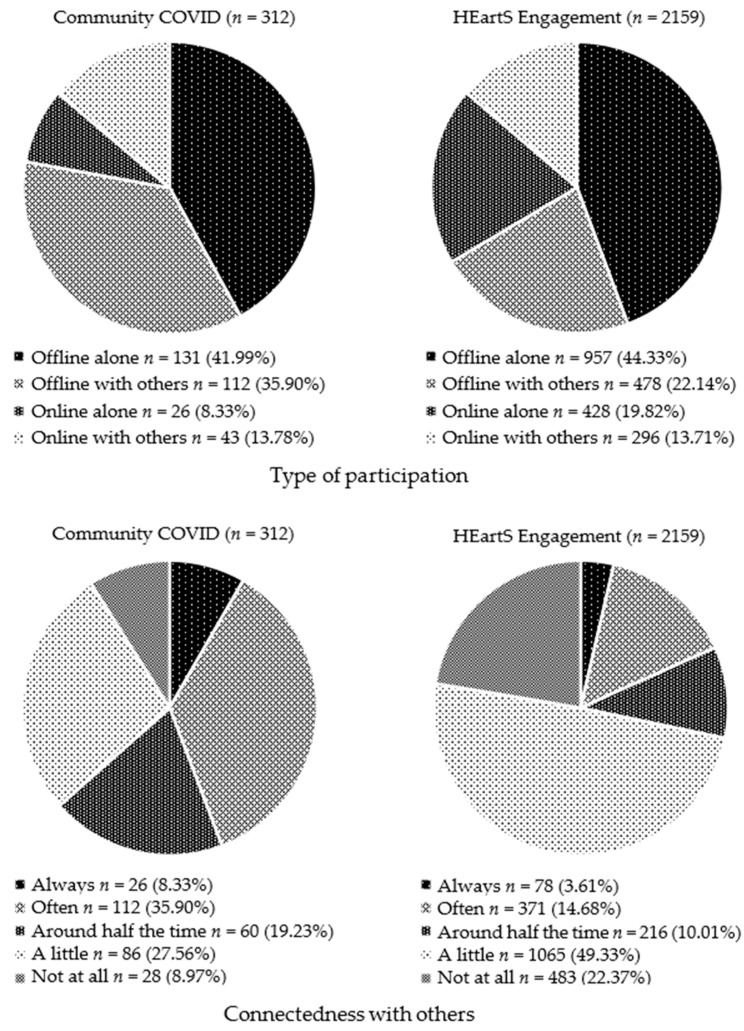
Both surveys: type of participation and connectedness with others.

**Figure 5 ijerph-20-06943-f005:**
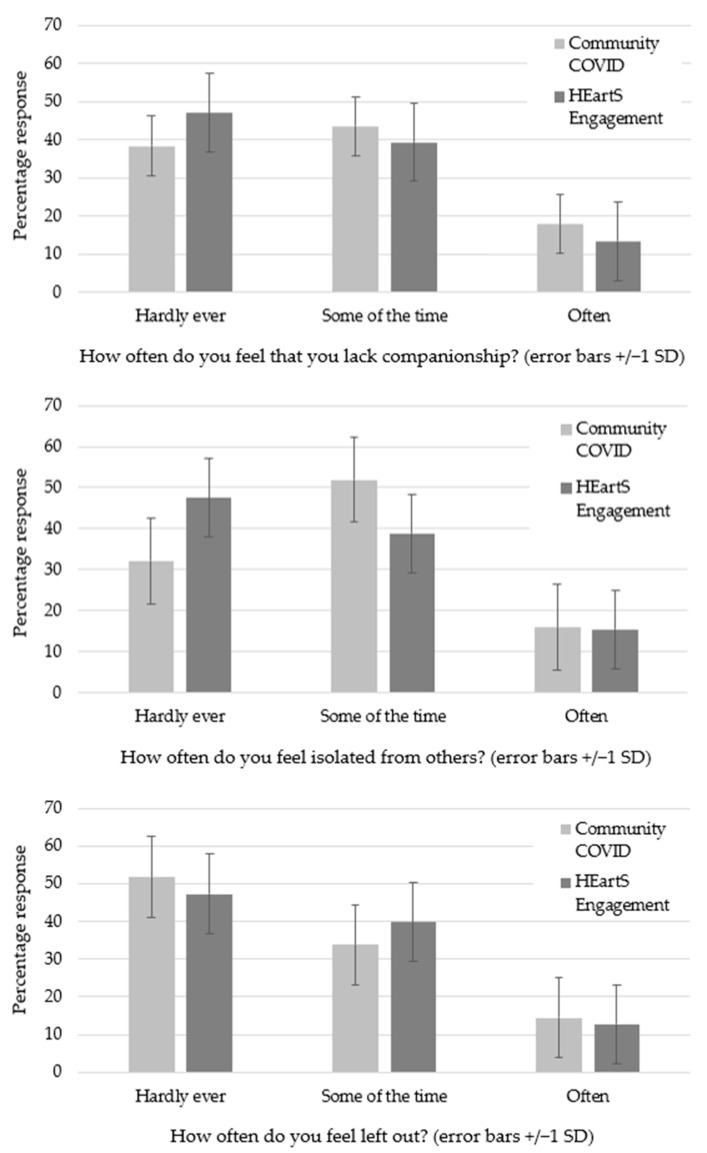
Both surveys: UCLA 3-item Loneliness Scale scores.

**Table 1 ijerph-20-06943-t001:** Community COVID: frequency and percentage of responses.

Question	Responses	Frequency	Percentage
How often have you taken part in these activities compared with before COVID-19 restrictions? (Please select one)	Much less often	14	4.49%
	Quite a lot less often	13	4.17%
	A little less often	13	4.17%
	No change	13	4.17%
	A little more often	66	21.15%
	Quite a lot more often	116	37.18%
	Much more often	77	24.68%
How do you usually take part in these activities? (Please select one)	Offline alone	130	41.67%
	Offline with other people	112	35.90%
	Online alone	26	8.33%
	Online with other people	43	13.78%
How much have these activities helped you feel connected with other people? (Please select one)	Not at all	30	9.62%
	A little	86	27.56%
	Around half the time	60	19.23%
	Often	110	35.26%
	Always	26	8.33%

**Table 2 ijerph-20-06943-t002:** Community COVID: loneliness and wellbeing means and SDs.

Measurement Scales	Items	Mean (SD)
Please select one box (out of five: 1 = I don’t feel; 2 = I feel a little bit; 3 = I feel fairly; I feel quite a bit;5 = I feel extremely) for each word to indicate how you feel on a typical day. (UCL Museum Wellbeing Measure: range 6–30; total score ≥ 18 indicates high wellbeing)	Active	3.03 (1.12)
	Alert	3.26 (1.08)
	Enthusiastic	3.12 (1.13)
	Excited	2.39 (1.17)
	Happy	3.06 (1.12)
	Inspired	3.12 (1.10)
	Total wellbeing (out of 30)	17.98 (6.72)
Please select one box (out of three: 1 = Hardly ever;2 = Some of the time; 3 = Often) for each question to indicate how you feel on a typical day. (UCLA 3-Item Loneliness Scale: range 3–9; total score ≥ 6 indicates high loneliness)	How often do you feel that you lack companionship?	1.79 (0.72)
	How often do you feel isolated from others?	1.84 (0.68)
	How often do you feel left out?	1.63 (0.72)
	Total loneliness (out of 9)	5.27 (1.80)

**Table 3 ijerph-20-06943-t003:** Community COVID: unstandardized (B) and standardized (Std. Error B) regression coefficients.

Predictor Variables	B	Std. Error B	Beta	Sig.
Frequency of participation	0.38	0.16	0.11	0.020
Connectedness score	1.35	0.24	0.29	0.001
Type of participation—offline or online	−0.12	0.63	−0.01	0.979
Type of participation—alone or with others	−0.01	0.54	−0.001	0.880
Age range	0.27	0.17	0.08	0.110
Loneliness score	−1.21	0.15	−0.40	0.001

**Table 4 ijerph-20-06943-t004:** Both surveys: UCLA 3-item Loneliness Scale means and SDs.

	Mean (SD)		Significance	
Loneliness Scale Questions	Community COVID	HEartS Engagement	Two-Tailed Probability	Bonferroni Correction
How often do you feel that you lack companionship?	1.79 (0.72)	1.66 (0.70)	0.01 **	0.04 *
How often do you feel isolated from others?	1.84 (0.64)	1.70 (0.72)	0.02 *	0.08
How often do you feel left out?	1.63 (0.72)	1.65 (0.69)	0.95	3.80
Total loneliness (out of 9)	5.27 (1.80)	5.01 (1.88)	0.90	3.60

** *p* < 0.01 = highly significant; * *p* < 0.05 = significant.

## Data Availability

The data presented in this study for the smaller-scale survey are available upon request from the corresponding author, H.C. The data are not publicly available due to reasons of privacy. The data presented in this study for the larger-scale survey were obtained from the Centre for Performance Science, Royal College of Music, London, and the Faculty of Medicine, Imperial College London, and are available on request from author, A.W.

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
