# Peer review of "The Impact of Culture-, Health- and Nature-Based Engagement on Mitigating the Adverse Effects of Public Health Restrictions on Wellbeing, Social Connectedness and Loneliness during COVID-19: Quantitative Evidence from a Smaller- and Larger-Scale UK Survey"

_ijerph, 2023, doi:10.3390/ijerph20206943_

Round 1

Reviewer 1 Report

Dear Authors,

This manuscript presents a very interesting topic about the impact of cultural, health and nature-based engagement in reducing the detrimental effects of restrictions during The COVID. However, some information is needed to improve this manuscript. Please see the Comment/suggestion/ Questions below. Thank you and good luck.

Comment/Suggestion/Questions

1.       The title clearly describes the content of the manuscript.

2.     The abstract already contains the problems, objectives and methods used, and the results of the research. However, there are too many keywords. Suggestions should be no more than 5 keywords.

3.       The introduction has clearly described the theoretical background.

4.       The materials and methods.

Lines 299-300: The study used experimental and correlational designs, and univariate and multivariate methods to analyze data generated by from the smaller- and larger- scale surveys.

Questions: Would you like to explain why used experimental design?

Suggestion: In this manuscript, the researcher only discussed surveys, perhaps it is better not to include experimental design, even though experimental design was carried out in the research.

5.       The data and information in the manuscript are new and original.

6.       Findings and discussions are presented systematically and logically.

7.       The conclusion has clearly described the study.

Author Response

Responses to Reviewer 1

This manuscript presents a very interesting topic about the impact of cultural, health and nature-based engagement in reducing the detrimental effects of restrictions during The COVID. However, some information is needed to improve this manuscript. Please see the Comment/suggestion/ Questions below. Thank you and good luck.

Thank you for reviewing the manuscript. Please note that revisions are shown as Track Changes, Show Markup>Balloons>Show revisions in Balloons.

Comment/Suggestion/Questions

  1. The title clearly describes the content of the manuscript.

Thank you for your positive comment.

  1. The abstract already contains the problems, objectives and methods used, and the results of the research. However, there are too many keywords. Suggestions should be no more than 5 keywords

The number of keywords has been reduced to five, thank you for noting this.

  1. The introduction has clearly described the theoretical background.

Thank you for your encouraging comment.

  1. The materials and methods.

Lines 299-300: The study used experimental and correlational designs, and univariate and multivariate methods to analyze data generated by from the smaller- and larger- scale surveys.

Questions: Would you like to explain why used experimental design?

The sentence was written with reference to the type of analysis used for the quantitative survey data; it should possibly read ‘experimental and correlational analysis’, i.e., not ‘experimental and correlational design’; we agree with your suggestion below.

Suggestion: In this manuscript, the researcher only discussed surveys, perhaps it is better not to include experimental design, even though experimental design was carried out in the research.

We agree with your suggestion and have removed ‘The study used experimental and correlational designs…’.  Please see new lines 444-445. Thank you for noting that there was an issue here.

  1. The data and information in the manuscript are new and original.

Thank you.

  1. Findings and discussions are presented systematically and logically.

Thank you.

Author Response

Responses to Reviewer 2

The manuscript examines the role of culture, health and nature-based engagement in mitigating the adverse effects of lockdown restrictions on people’s mental health. The analysis of the two surveys indicates the significant impact of frequency of participation, connectedness with others, and loneliness on personal well-being. The comparison of the results from the two surveys demonstrates cost effectiveness of using a small survey to achieve similar results.

Thank you for reviewing the manuscript. Please note that revisions are shown as Track Changes, Show Markup>Balloons>Show revisions in Balloons.

I commend the authors’ effort in reviewing the existing literature on the impact of lockdown restrictions. However, since the authors are mainly concerned with online and offline activities and their impact on individual psychological well-being, the theoretical rationale underpinning the impact of these activities should be offered. The role of cultural impact needs to be defined and the literature on the concept reviewed.

Though there is little relevant literature on the rationale underpinning online and offline activities and their impact on individual wellbeing during COVID-19, material from two pre-pandemic articles, and the theories on which they draw, has been added from (old line 294) new lines 406-435

We have defined the role of cultural impact and reviewed four pertinent examples of the literature on the concept, after the sentence that starts ‘Of the large-scale surveys outlined above…’ (old line 275 onwards) new lines 377-389.

The additional material included here re: online and offline activities and cultural impact is referred to again in the Discussion section, (old lines 584-586 and 623-627) new lines 830-832 and 871-874.

In addition, I have concerns with the definition for the constructs in the research model. First, the variables in the model are not conceptually defined. For example, how is Participation offline or online different from Participation alone or with others? These two concepts could overlap with each other.

We identified that there could be four separate ways in which respondents participated in activities: offline with other people; offline alone; online with other people; and online alone; these options have been expressed more clearly on (old line 290) new lines 403-404, and are illustrated in the figure below (though we do not fee that it is necessary to include this figure in the manuscript):

Type of participation

             Online or offline

ONLINE

OFFLINE

Alone or with other people

ALONE

1) Online alone

2) Offline alone

WITH OTHER PEOPLE

3) Online with other people

4) Online alone

Table A2 (page 21) lists three types of activities: culture, health and nature-based engagement (C), participatory arts (P), and receptive arts (R). There is also a list of activities that are excluded from the HEartS survey. What is the definition for each category? It is unclear how “Dog talking looking after pets” is considered a “C,” not a “P.” Why is “Practicing/performing dance” a “P,” not a “C”? It is also unclear as to why “Working, studying and training courses” is excluded, while “listening to audio books/podcasts” is a “R.” Each type of activities shall be clearly defined.

The use of the initials in Tables A2 and A3 was more to explore the range of different types of activity during the research process and we quite agree, the categories are open to interpretation and probably don not serve any useful purpose as far as the reader is concerned. Consequently, we have removed these initials from the tables. Please see new lines 1006-1008 and 1013-1015.

Methodologically, it is unclear how these constructs (e.g., frequency of participation is calculated, connectedness with others) are measured. UCLA has twenty measurement items; however, the manuscript has only three. What is the museum wellbeing measure? The authors state that the measure scales range from 6-30, but all the measurement items have a mean of lower than four (Table 2). The validity and reliability of all constructs are not examined.

Thank you for your methodological comments which are addressed in turn below: 

Frequency of participation (old lines 359-361) new lines 520-525, Question 4: “‘How often have you taken part in these activities compared with before COVID-19 restrictions?’ Respondents ticked one box and frequency and percentage of response were calculated (Table 1); 83 per cent of respondents took part more often (a little more 21%; quite a lot more: 37%; much more: 25%)”. We have clarified how this construct has been calculated. The lines now read ‘Respondents ticked one box from the seven possible responses (Table 1) where frequency refers to how frequently each response occurred i.e., the number of respondents who selected each response with the percentage of those respondents calculated out of the total number of respondents; 83 per cent of respondents took part more often (sum of a little more: 21%; quite a lot more: 37%; much more: 25%).

Connectedness with others (lines 366-369) new lines5 30-533 and 534-537, we have added notes to Questions 5 and 6 (connectedness) to say that they were calculated in the same way as Question 4.

UCLA: (Old line 380) new line 561 states that Respondents completed the UCLA 3-item Loneliness Scale; this is not the version with 20 items; it is cited on new line 458 under Materials and Methods and the citation (26) is given in full in the References section, new lines 1143-1144. Please also see https://www.icmha.org/wp-content/uploads/2020/02/UCLA-Loneliness-Scale.pdf which explains (p 19) that the longer scale was shortened to three questions in 2004, first tested with over 2,100 adults it was found to be a valid and reliable measure of loneliness.

Museum Wellbeing Measure (old line 371) is a measure that has been in use for 10 years. References 47 and 48 in the References section, new lines 1139-1142, provide the citations for this measure, which is cited under Materials and Methods on new lines 461-464. The articles show validity and reliability of the scale. Please also see: https://www.ucl.ac.uk/biosciences/culture-nature-health-research/ucl-creative-wellbeing-measures and: Thomson, L.J. and Chatterjee, H.J. (2022). Heritage, Creativity, and Wellbeing: Approaches for Evaluating the Impact of Cultural Participation; Using the UCL Museum Wellbeing Measures. In P. Everill & K. Burnell (Eds), Archaeology, Heritage, and Wellbeing: Authentic, powerful, and therapeutic engagement with the past, Ch. 4, Abingdon: Routledge. https://www.routledge.com/Archaeology-Heritage-and-Wellbeing-Authentic-Powerful-and-Therapeutic/Everill-Burnell/p/book/9781032021652

With respect to your comment that the measure scales range from 6-30, but all the measurement items have a mean of lower than four (Table 2, new lines 569-623): The measures consist of 6 items that can each score between 1 and 5 consequently the lowest possible score is 6 (6 x 1) and the highest possible score is 30 (6 x 5). Table 2 shows the scores for each of the items separately, these scores can be between 1 and 5, and the mean scores are between 1 and 5, in fact they are between 2 and 4. Further explanation for Question 7 has been provided to add clarity here (old lines 371-372) new lines 544-552.

Regarding your comment about the validity and reliability of all constructs, internal consistency reliability for the UCL Museum Wellbeing (Thomson and Chatterjee 2014) was established using Cronbach’s alpha producing an alpha level of 0.81, where scales of 0.7 and over are regarded as reliable (Brace et al. 2009). The UCLA 3-Item Loneliness Scale has been repeatedly demonstrated to display satisfactory internal consistency, test-retest reliability, and concurrent and discriminant validity [Hughes et al 2004)], and its three-item short-form version is currently the recommended loneliness measure at the national level in the UK (Office for National Statistics 2018). Regression analysis was used to ascertain criterion predictive validity for analysis of wellbeing, loneliness variables and the constructs in Table 1 (how often respondents take part, how they take part, and how connected they feel) which examined the extent to which each measure predicted other measures. This information has been outlined in the text in connection with all constructs (from old line 313) new lines 461-473.

The authors claim that the approximation of the results from the two surveys could justify using a small size survey for a large one. I am not sure that the statement is valid unless the representativeness is established. In addition, there is no significance test comparing the difference in the statistical analysis of the two surveys.

The authors of the larger survey feel that a direct comparison between the two surveys is not appropriate for several reasons: the larger survey included a greater number and range of questions than the smaller survey, it was part of a series of HEartS surveys started prior to COVID, that participants were paid to complete the larger survey whereas for the smaller survey, participation was entirely voluntary (see Strengths and Limitations, old lines 590-607, new lines 886-906), different sampling methods were used (Community COVID used a convenience sample whereas HEartS used quotas), and it is probably not statistically reliable to compare just a subset of the larger study with all of the smaller study. Also, and as pointed out in (old lines 484-485) new lines 741-743 ‘As HEartS Engagement did not measure wellbeing directly, correlations with loneliness could not be carried out’. We have consequently removed the claim that the approximation of the results from the two surveys could justify using a small size survey for a large one, as you suggest: New lines 29-31 have therefore been removed from the Abstract and (old lines 623-628) new lines 871-874 have been removed from the end of the Conclusions. Old lines 587-591 have been moved from the Discussion (and re-phased slightly) to act as an ending to the Abstract and the Conclusions, new lines 875-878.

In terms of the quality of writing, while overall understandable, the manuscript could use some polish. There are some confusing lines (e.g., line 17, line 251). There is an error in line 476.

Parts of the manuscript have been re-phrased (please see track changes). Thank you for pointing out confusing lines, addressed in turn:

Lines 15-18 read: ‘Fewer surveys examined ways in which adverse effects of public health restrictions might be alleviated. The purpose of the current study was to assess the impact of culture, health and nature-based engagement on mitigating adverse effects of restrictions such as lockdown, shielding and social distancing’. The sentence to which you refer was written as briefly as possible to save words in the word count for the abstract. This sentence and the one before it have both been re-phrased to improve clarity and now read: ‘Fewer surveys examined ways in which adverse effects of public health restrictions, such as lockdown, shielding and social distancing, might be alleviated. The purpose of the current study was to assess whether culture, health and nature-based engagement would mitigate the effects of these restrictions on psychological wellbeing, social connectedness and loneliness’.

Line 251 (now new line 338) is the last line of the sentence: ‘The study assessed associations between frequency of receptive engagement (visits to cinemas; museums; galleries etc.) and loneliness in older adults using the three-item Revised UCLA Loneliness Scale [26] and found a negative correlation between receptive arts and loneliness; engagement every few months was associated with lower levels of loneliness’. The last line has been rephrased to add clarity: ‘‘The study assessed associations between frequency of receptive engagement (visits to cinemas; museums; galleries etc.) and loneliness in older adults using the three-item Revised UCLA Loneliness Scale [26] and found a negative correlation between receptive engagement and loneliness; visiting cultural venues every few months appeared to reduce loneliness’. [As we are reporting a correlation, we cannot state that there was cause and affect so we have used the words ‘appeared to’]

Old lines 475-478 (now new lines 680-682) read: ‘In line with the hypothesis, increased levels of connectedness were significantly associated with higher scores on the UCL Wellbeing Measure; respondents who felt more connected with others reported better psychological wellbeing than those who felt less connected’. We are not sure when you say an error in old line 476 if you mean an error in interpreting the statistics or a typo. As new lines 412-415 read: ‘Regression coefficients for the predictor variables entered into the model (Table 3) showed that connectedness was a highly significant predictor with a positive relationship to wellbeing, where increased connectedness was associated with increased wellbeing (p<.001)’, in statistical terms, this sentence appears correct. We have not spotted any typos here.

Reviewer 3 Report

I am not sure why you used two datasets. Although you explain that you added the questions to a larger survey, I think this requires further explanation. I think you stated the surveys were held at the same time, but again, it is not clear. It appears there were significant differences in their results but analyzing those differences was not the purpose of the study. Did you consider using only one dataset?

I may be wrong based on the style required by the journal, but the text within your tables can be difficult to follow since everything is centered and there are several lines for some statements. Unless this is a requirement, consider reformatting the tables and perhaps reduce the font size.

Figure 4 needs to be reformatted. The call out boxes are so big that the pie charts are overshadowed. This should be something one can look at and easily understand, but it is unclear. Consider adding a legend-table with the data under the pie charts.

I am impressed with the amount of information in the article. After a thorough reading, however, I am not clear if you made the intended points to support the purpose statement in the abstract –

 The purpose of the current study was to assess the impact of culture, health and  nature-based engagement on mitigating adverse effects of restrictions such as lockdown, shielding and social distancing.

I do not see a substantial amount of specific analyses about nature-based engagement although it is in the title and purpose statement. Please highlight those analyses.

It seems that a lot of the analysis and discussion sections are about comparing the results of the two surveys but not clearly supporting the answers to your purpose statement, aims, or hypotheses.

This is well written but perhaps too dense. Some of the information in the introduction may not be necessary. The discussion seems to continue reporting the results rather than discussing the overall findings and purpose of the study.

Author Response

Thank you for reviewing the manuscript. Please note that revisions are shown as Track Changes, Show Markup>Balloons>Show revisions in Balloons.

I am not sure why you used two datasets. Although you explain that you added the questions to a larger survey, I think this requires further explanation. I think you stated the surveys were held at the same time, but again, it is not clear. It appears there were significant differences in their results but analyzing those differences was not the purpose of the study. Did you consider using only one dataset?

Under Materials and Methods, (from old line 303) new lines 445-448 read ‘To increase the reach of data collection, questions were embedded in the larger scale ‘Health, Economic, and Social Impact of the Arts’ Phase 2 Engagement survey’. We have now added further explanation in terms of reaching not only more respondents but a wider cross section of respondents as the larger survey used quota sampling. Thank you for noting that this aspect was not clear.

Old lines 335-336 (now new lines 501-502) provide the dates for both surveys ‘Community COVID for 12 weeks (07 Sept-27 Nov 2020) and HEartS Engagement for four weeks (05-30 Oct 2020)’ [Apologies for the wrong end date for the 12-week survey in the text, now amended]. This means that the smaller survey was started four weeks before the larger survey and ended four weeks after it. The larger survey was able to take place more quickly because its administrators already had participants from the previous survey available to participate who were paid for their participation (through Qualtrics). The smaller survey took longer to recruit voluntary (and anonymous) respondents contacted via project partners’ mailing lists.

Analysing the differences between the two surveys was not the purpose of the study because we were not comparing like with like; as we have explained to Reviewer 2, the authors of the larger survey felt that a direct comparison between the two surveys was not appropriate for several reasons: the larger survey included a greater number and range of questions than the smaller survey, it was part of a series of HEartS surveys started prior to COVID, that participants were paid to complete the larger survey whereas for the smaller survey, participation was entirely voluntary (see Strengths and Limitations, old lines 590-607, now new lines 833-853) and different sampling methods were used (Community COVID used a convenience sample whereas HEartS used quotas). Also, and pointed out in (old lines 484-485) new lines 689-690: ‘As HEartS Engagement did not measure wellbeing directly, correlations with loneliness could not be carried out’.

In response to whether we considered using only one dataset, we thought that we would be able to use one data set when we initially applied for funding in that we had sufficient partners to obtain a large sample of anonymous participants by advertising the survey through their mailing lists/newsletters, etc. Further on into the research our partners seemed less confident about the numbers about the numbers of participants they might attract as it became clear that numerous surveys were being carried out during COVID-19.  We consulted with our collaborators at Imperial College/Royal College of Music about how they obtained participants for their surveys, and they informed us that they did this through Qualtrics which involved payments to this company and to the participants. As we had not obtained sufficient funding to cover the cost of such a survey through Qualtrics or similar organisation, we opted to take up the IC/RCM offer of embedding a subset of our questions (areas not covered by HEartS) into their survey and, also, analysing the data from questions which were similar to those in the smaller survey. Before sending out the smaller survey we rewrote some of the questions to use the same wording as the larger survey. We did not consider, however that the level of detail we have just described here about why we decided to use two surveys was necessarily relevant to include in the paper, other than to state it was to increase the reach of the smaller survey.

Furthermore, other studies have used data sets from two surveys; and example is cited from (old line 254) new line 345 onwards: ‘In a survey determining multidisciplinary priorities for mental health research, over 2,000 respondents, contacted through a mental health charity mailing list of stakeholders with lived experience, completed an online survey during the first five weeks of lockdown (Mar-Apr 2020). Respondents were asked for their two biggest concerns and their coping strategies for maintaining health and wellbeing [25]. Though the survey was biased in having four times as many females as male respondents and fewer young people under 18, it was complemented by the more representative public Ipsos MORI’s Omnibus survey’. A further example (though not included in the current study as it occurred pre-COVID) is Fujiwara et al. (2014) who quantified the impacts of culture and sport, based on data sets from the surveys ‘Understanding Society Wave 2’ and the ‘British Household Panel Survey’ and showed that, after controlling for other determinants (e.g., income, gender, health, region, and marital and employment status), engagement with the arts, visits to museums, libraries and heritage sites, and sports participation were associated with higher wellbeing.

I may be wrong based on the style required by the journal, but the text within your tables can be difficult to follow since everything is centered and there are several lines for some statements. Unless this is a requirement, consider reformatting the tables and perhaps reduce the font size.

The Vancouver style guide reads that text in tables should be centred and several recent articles with tables display the text in this way. We have also seen the text ranged left and the numeric information centred in recent articles, so we have ranged the columns of text left to improve readability, leaving the numbers centred and made the text in the tables one point size smaller, as you suggest.

Figure 4 needs to be reformatted. The call out boxes are so big that the pie charts are overshadowed. This should be something one can look at and easily understand, but it is unclear. Consider adding a legend-table with the data under the pie charts.

Thank you for your suggestion; Figure 4 has been reformatted by adding a legend with the data under the pie charts which appears to make the figure easier to understand.

I am impressed with the amount of information in the article. After a thorough reading, however, I am not clear if you made the intended points to support the purpose statement in the abstract –

The purpose of the current study was to assess the impact of culture, health and nature-based engagement on mitigating adverse effects of restrictions such as lockdown, shielding and social distancing.

I do not see a substantial amount of specific analyses about nature-based engagement although it is in the title and purpose statement. Please highlight those analyses.

The analyses were about engagement with activities that respondents participated in during COVID-19 taken as a whole and which included cultural activities (e.g., music, art), healthy activities (e.g., walking, sports, and fitness) and nature-based activities (e.g., gardening, visits to natural environments). Question 1 (see Appendix 1) asked ‘What have you done with your free time during COVID-19 restrictions? Please list your main activities below: (These could include arts, crafts, choirs, dance, DIY, fit-ness, games, gardening, music, nature, puzzles, reading, volunteering, walking, writing, etc.)’. Many of these activities were a combination of more than one type of engagement (e.g., art or walking in a natural environment; fitness obtained through gardening or DIY), Please see point made in Discussion, new lines 759-763. We did not analyse nature-based activities separately from other areas of engagement (neither did we analyse culture- or health-based engagement separately). Instead, we considered the frequency and type of participation of these activities and their relationship with improved wellbeing and reduced loneliness (see Table 3 predictor variables). To clarify that we looked at engagement as a whole we have amended the title and subsequent uses of the types of engagement by placing hyphens after ‘culture’ and ‘heath’ (so it is clear that we are considering culture-based, health-based, and nature-based engagement, not just that which is nature-based), as in: The Impact of Culture-, Health- and Nature-based Engagement on Mitigating the Adverse Effects of Public Health Restrictions on Wellbeing, Social Connectedness and Loneliness during COVID-19: Quantitative Evidence from a Smaller- and Larger-scale UK Survey’.

Although we did not analyse nature-based engagement separately, we appreciate the point you are making about the separate types of engagement, so in the Discussion, we have considered health-based engagement (new line 761), culture- and nature-based engagement, new lines 747, 759, and nature-based engagement, new lines 761-773.

It seems that a lot of the analysis and discussion sections are about comparing the results of the two surveys but not clearly supporting the answers to your purpose statement, aims, or hypotheses.

As outlined in the Abstract, the purpose of the study was to ascertain whether participation in activities during COVID-19 would mitigate the effects of public health restrictions. The study analysed frequency of participation and how respondents participated, i.e., online or offline and with or without other people, in addition to measuring wellbeing and loneliness to achieve this purpose. We have revised the Discussion to support more clearly the answers to the purpose (new lines 681-684), aims and hypothesis.

This is well written but perhaps too dense. Some of the information in the introduction may not be necessary. The discussion seems to continue reporting the results rather than discussing the overall findings and purpose of the study.

We have removed some of the information from the Introduction where it does not seem entirely necessary, please see track changes (though have added other material in line with Reviewer 2).

We have re-written parts of the Discussion in favour of discussing the overall findings and purpose of the study, please see track changes.

Round 2

Reviewer 2 Report

Please see the attached for comments

Author Response

ijerph-2452339 Responses to Reviewer 2 Second Round

Review: The Impact of Culture-, Health- and Nature-based Engagement 2 on Mitigating the Adverse Effects of Public Health Restrictions 3 on Wellbeing, Social Connectedness and Loneliness during 4 COVID-19: Quantitative Evidence from a Smaller- and Larger-scale UK Survey

While I commend the authors’ diligent effort in revising the manuscript, some of comments were not addressed. Below is the screenshot.

I commend the authors’ effort in reviewing the existing literature on the impact of lockdown restrictions. However, since the authors are mainly concerned with online and offline activities and their impact on individual psychological well-being, the theoretical rationale underpinning the impact of these activities should be offered. The role of cultural impact needs to be defined and the literature on the concept reviewed.

In addition, I have concerns with the definition for the constructs in the research model. First, the variables in the model are not conceptually defined. For example, how is Participation offline or online different from Participation alone or with others? These two concepts could overlap with each other.

After a careful review of the manuscript, I feel that my previous comments are still valid. I’d like to build on the comments and elaborate on the areas of improvement.

Thank you for taking the time to build on your comments and elaborate areas of improvement. Please note that all line numbers refer to the revised version of the manuscript as submitted on this occasion. As revisions are shown in ‘Inline’ under Show Markup, the line numbers are increased; line numbers given below reflect this new numbering.

Throughout the manuscript, the authors mentioned culture, and never defined culture. Is culture local confined to a group of individuals (e.g., puts, darts and gardening)? Is culture race or region specific? Is cultural impact (line 223-235) the same as the impact of health and nature-based engagement? A clear definition for each category of activities is needed.

The reference to pubs, darts and gardening (line 221) is taken directly from the DCMS where they suggest that these activities are not cultural, contrasting ‘culture’ with ‘other activities’, as in: “About half of the respondents had low levels of engagement with culture but took part in other activities such as ‘pubs, darts and gardening’, and 11 per cent were ‘detached from mainstream pastimes and social events outside of television’ [28] (p. 169)”. Prior to the discussion of cultural impact (lines 292-301) we have added the DCMS (2016) definition of culture from their publication ‘The Cultural White paper’ (lines 286-291).

We have defined ‘impact’ (lines 291-292) and ‘cultural impact’ (lines 293-301). Cultural impact is not seen as exactly the same as health or nature-based impact although these do align with the more general definition of impact. To provide a clear definition of each category, as you suggest, health impact is now defined on lines 301-306, and nature-based impact is now defined on lines 306-310.

Table A3 lists 20 categories of activities. Which category is culture based, health related, or nature-based? The authors categorize four types of participation: online alone, online with others, offline alone, and offline with others. While this categorization applies in most of the activities, they don’t seem to fit some others including gardening/looking after plants, visiting natural environments, dog walking/looking after pets, DIY/home improvements, crafts, textiles or decorative arts, and practicing, performing dancing.

For the four types of activity, participants in the surveys determined their responses as to how they had participated (online alone, online with others, offline alone, and offline with others) in the activities listed in Table A3 i.e., this information was self-reported (so presumably participants who responded had no issues in allocating one of the four categories of participation to the activities they selected). However, as expressed in line 516, 1488 participants selected ‘not applicable’ so had not taken part in these particular activities. To stress that the categories were selected by participants the point has been emphasised, lines 511-513.

[The examples of activities you suggest that do not seem to fit with these four categories are addressed in turn in terms of the possible rationale that enabled participants to respond:

  • Gardening/looking after plants: Most participants carried out these activities offline and alone though offline with others could have applied to allotments, online alone could be viewing online material, and carrying out an activity simultaneously, such as potting a cutting or planting seeds, and online with others could have been attending a gardening club on Zoom.
  • Visiting natural environments: Most participants did this offline with others who might have included members of their household (permissible during COVID) and there may have been security issues involved with visiting these environments alone. Whereas online visits are in the minority, it might have been that participants were reporting viewing online educational/documentary material of natural environments.
  • Dog walking/looking after pets: Again, mostly offline, and roughly equally with and without others (the former may apply to small groups of dog walkers with social distancing between them). It is possible that there was online material about looking after pets available, particularly during lockdown.
  • DIY/home improvements: Most participants reported doing this offline and alone, others may have included help from family members, and online activities could have included instructional material.
  • Crafts, textiles or decorative arts: Mostly offline alone, these activities could have been carried out with other household members. For online activities, several community organisations to our knowledge sent out art materials packs to support classes which changed from in-person to online sessions during COVID.
  • Practicing, performing dance/play etc. Similar with the other activities, taking part in these activities could have been moved from in-person to online during the pandemic. Certainly, this appeared true of online choirs.]

In the Discussion section, the authors use the word hypothesis (see the screenshot below). However, there are no clear hypotheses. Neither are there supporting arguments that are based on any theoretical frameworks. It is unclear how participation offline or online is a variable (Table 3). What is the definition for participation alone or with others? What is the theoretical argument supporting its impact on wellbeing?

The theoretical framework for the study, self-determination theory (Deci & Ryan 2000) has been described towards the end of the Introduction (page 7, lines 314-318 and lines 348-354), mentioned in the Abstract (lines 16-17), in the Discussion (lines 583-584 and lines 592-600), and in the Conclusions (line 749). In the introduction and discussion (lines as above) the links of this theoretical supporting argument with wellbeing have been expressed. We have clarified the hypotheses, lines 348-356.

In Table 3, the variables ‘participation offline or offline’ and ‘participation alone or with others’ are dichotomous categorical data (please see lines 487-491) which are acceptable in regression analysis as predictor variables (though not as criterion variables). In the analysis, frequency of participation had seven levels, age range had seven levels, connectedness had five levels, loneliness had four levels, participation offline or offline had two levels, and participation alone or with others had two levels of each variable. Had either type of participation variable been statistically significant, then further analysis would have been carried out to determine the direction of any effects e.g., whether offline or online activity was a better predictor of wellbeing. To make Table 3 clearer for readers, the variables have been expressed a little more fully.

The definition for participation alone or with others has been included in the Introduction shortly after the definitions of connectedness and loneliness on lines 331-334.

  1. Discussion

The aims of the current study were to establish how participants spent their free time under lockdown and other public health restrictions, and evaluate the effects of these activities. The objectives were to analyze data from both surveys to assess the frequency of activities and determine how these were carried out.). The purpose of the study was to assess whether culture-, health- and nature-based engagement during COVID-19 would mitigate the effects of public health restrictions on psychological wellbeing, social connectedness and loneliness. In line with the hypothesis, respondents who felt more connected with others reported better psychological wellbeing than those who felt less connected.

Methodologically, the authors need to indicate the source for the list of activities.

For Community COVID, participants were asked to generate and type onto the online form which activities they took part in most (lines 419-421 under Question 1). As explained, these activities were grouped into 30 categories using similar wording wherever possible to the HEArts Survey (Tymoszuk et al. 2021) carried out prior to COVID, referred to on lines 421-424. We have added the citation [33] for this survey under Question 1 (line 423) to indicate that the wording came from this previous research. As the list of activities for Community COVID was generated by participants and categorised as part of the analysis, this process has been described under 31.1 Analysis of the Community COVID survey, rather than under 2. Materials and Methods.

It seems that while there are a lot of graphs, the meaning behind the graphs is unclear. For example, Figure 6 shows loneliness across regions. What does that mean to the proposed relationships? What is the purpose of the second part of Figure 4? The authors need to clearly state their research question(s) and present only relevant info. in the manuscript.

We agree with your observation regarding Figure 6 and allied text under 3.2.3 Loneliness across regions; the graphs were part of the process of the researchers deciding how similar the smaller and the larger surveys were to each other, but we agree that this line of enquiry is no longer as relevant as it might be, adding little to the proposed relationships, consequently we have removed this section and Figure 6 (page 16, lines 552-564). We have also removed ‘and region’ from line 30 of the Abstract.

The second part of Figure 4 illustrates the level of connectedness with others across both surveys, where (for Question 6) participants were asked to rate the extent to which they felt connected with others. Lines 494-496 explain the importance of connectedness ‘connectedness was a highly significant predictor with a positive relationship to wellbeing, where increased connectedness was associated with increased wellbeing (p<.001)’. Unless the second part of Figure 4 is illustrated, there is nowhere else in the text that reports the HEartS Engagement survey findings for levels of connectedness. The whole of Figure 4 could be expressed instead as a table without the pie charts, but we felt that these allowed for more easily assimilated graphic comparison of the two data sets (from the smaller and larger surveys).

Btw, the authors need to check the percentage of offline alone number for the HEartS Engagement survey (p. 12)

New page 14 (Figure 4). Thank you for pointing out this typo; corrections have been made to this figure.

Hope that the comments above are helpful for the authors to further improve the manuscript.

Thank you for your helpful comments.